# Research on Identification of Minimum Parameter Set in Robot Dynamics and Excitation Strategy

**DOI:** 10.3390/s25185749

**Published:** 2025-09-15

**Authors:** Zhiqiang Wang, Jianhai Han, Xiangpan Li, Bingjing Guo, Lewei Lu

**Affiliations:** 1School of Mechatronics Engineering, Henan University of Science and Technology, Luoyang 471003, China; wzq1992@stu.haust.edu.cn (Z.W.); xiangpanli@haust.edu.cn (X.L.) bingjing@haust.edu.cn (B.G.); 2Henan Provincial Key Laboratory of Robotics and Intelligent Systems, Luoyang 471003, China; 3School of Materials Science and Engineering, Henan University of Science and Technology, Luoyang 471000, China; lulewei@163.com

**Keywords:** robot dynamic, parameter identification, screw theory, linear matrix equation, least squares method

## Abstract

The minimal parameter set is fundamental to robot dynamic identification, enabling efficient and identifiable modeling for control and simulation. In this paper, the Newton–Euler method is employed to formulate the robot dynamics. By leveraging screw theory, the model is expressed in a matrix form that is linear with respect to the robot’s inertial parameters. The Kronecker product is then applied to transform the matrix equation into an equivalent vector–matrix representation. Subsequently, full-rank decomposition is used to reduce the dimensionality of the parameter vector, resulting in the minimal dynamic parameter set of the robot. Following this, excitation signals are sequentially applied to each joint, starting from the end-effector and progressing toward the base, enabling a stepwise identification of the minimal parameter set using the least-squares method. The identified minimal parameters are then incorporated into the mass matrix of the dynamic model, enabling the implementation of forward dynamic simulation. Experimental validation is conducted on a planar 3R robot. The results demonstrate that the sequential excitation strategy accurately identifies dynamic parameters while ensuring the robot’s safety. Furthermore, the forward dynamic simulation closely replicates the kinematic behavior of the actual robot.

## 1. Introduction

The dynamic model of a robot system plays a crucial role in both simulation and high-precision control [1]. For serial robotic systems, the identification of dynamic parameters fundamentally hinges on determining the minimal parameter set, which encapsulates all identifiable inertial and friction parameters required to predict joint torques [2,3,4]. The need for such a minimal set arises from the intrinsic kinematic and dynamic constraints inherent in the robot’s mechanical structure. These constraints lead to linear dependencies among the inertial parameters in the dynamic equations of motion, resulting in a rank-deficient regressor matrix during the formulation of the identification problem. Moreover, the minimal parameter set cannot be directly applied to forward dynamics, which limits its utility in simulation applications. Recently, reinforcement learning (RL) has shown great potential in robotic control. However, training agents through direct interaction with real robots is time-consuming and potentially unsafe [5,6,7]. By leveraging forward dynamics to simulate the real robot, the entire RL training process can be conducted in a simulated environment, enabling the agent to perform extensive trial-and-error learning. This approach enables the simulation of robot motion and dynamic response under various complex operating conditions, effectively avoiding risks associated with physical interactions, reducing training costs and time, and accelerating model iteration and optimization.

Dynamic models of robotic systems can be formulated using either the Lagrangian or Newton–Euler method, both of which are mathematically equivalent. However, the recursive structure of the Newton–Euler formulation is more suitable for numerical implementation. By employing screw theory, the dynamics of each link can be expressed as a matrix form, thereby transforming the overall dynamic model into a linear matrix equation with respect to the spatial inertia matrices [8,9,10,11]. Fu et al. [10] proposed a method that employs the Kronecker product to convert such linear matrix equations into equivalent vector–matrix forms. By extracting linearly independent columns from the coefficient matrix, the dimensionality of the parameter vector is reduced, yielding the minimal dynamic parameter set. It is worth noting that the Kronecker product and vectorization techniques are applicable to a wide range of linear matrix equations, including but not limited to AXB + CXD = E and AXB = C [12,13,14,15]. The vector–matrix and matrix formulations are mathematically equivalent and yield identical solutions.

In the context of forward dynamics, the mass matrix itself can also be expressed as a linear matrix equation with respect to the spatial inertia matrices. Therefore, the vectorization technique is equally applicable for computing the mass matrix. With the identified minimal parameter set, the mass matrix can be reconstructed and used in conjunction with the inverse dynamics vector–matrix equation to establish a forward dynamic model. In this way, both forward and inverse dynamic models can be derived from the minimal parameter set, thereby extending the applicability of the identified dynamic model beyond torque prediction.

In robotic dynamic parameter identification, excitation signals must be simple, reproducible, and continuously differentiable [16]. Consequently, finite Fourier series have been widely adopted in robotic parameter identification tasks [17,18,19,20]. By minimizing the condition number of the regressor matrix while incorporating constraints on joint positions and velocities, the sensitivity of parameter estimation to measurement noise can be reduced. Tika et al. [19] employed a memetic algorithm to optimize the excitation trajectory, achieving a low condition number and small standard deviations in parameter estimates. Qin et al. [21] used a genetic algorithm to optimize the condition number excitation deriving Fourier series-based excitation trajectories for parameter identification. Nonlinear optimization methods can also be utilized to improve the condition number and generate optimal excitation trajectories. However, the choice of constraints significantly affects the optimization outcome, and these constraints vary across different experimental setups.

In summary, methods that aim to minimize the condition number of the regressor matrix to generate optimal excitation trajectories require numerical optimization, which can be computationally expensive and exhibit variable convergence behavior across trials. Furthermore, the condition number is highly sensitive to joint positions, velocities, and accelerations, and trajectories that minimize it may involve aggressive motions that compromise operational safety. To ensure safety and practical implementation, sinusoidal functions are adopted as excitation trajectories, with only one joint actuated at a time, enabling a single-input multiple-output (SIMO) identification scheme. By sequentially exciting joints from the end-effector toward the base, the complete minimal parameter set is identified in a structured and reliable manner.

The remainder of this paper is organized as follows. Section 2 introduces the foundational materials and methods, including the homogeneous transformation of twists and wrenches based on screw theory, and presents the robot dynamic modeling framework using the Newton–Euler method. Section 3 focuses on the derivation of the minimal dynamic parameter set: it first reformulates the recursive dynamics into a block matrix representation, then applies Kronecker product-based linearization to transform the matrix equation into a vector form, and finally extracts the minimal parameter set via full-rank decomposition. The section concludes with the application of the identified parameters in forward dynamics modeling. Section 4 details the sequential excitation strategy, designed to ensure operational safety and persistent excitation by actuating one joint at a time. Experimental validation and parameter identification results on a planar 3R robot are presented in Section 5, demonstrating the accuracy and effectiveness of the proposed approach. Finally, Section 6 provides concluding remarks and discusses potential extensions to higher-degree-of-freedom robotic systems.

## 2. Modeling and Methodology

### 2.1. Homogeneous Transformation of Twists and Wrenches

A velocity twist is a six-dimensional vector composed of angular velocity ω and linear velocity v, defined as V=ωTvTT. As illustrated in Figure 1, the transformation of the velocity twist from link frame i to frame i−1 is given by:(1)Vi−1=AdTi−1,iVi,
where Vi and Vi−1 denote the velocity twists expressed in link frames i and i−1, respectively; AdTi−1,i∈ℝ6×6 is the adjoint transformation matrix associated with the homogeneous transformation matrix Ti−1,i, which represents the transformation from frame i to frame i−1.

For an arbitrary homogeneous transformation matrix T, the adjoint transformation matrix is defined as:(2)$${\mathrm{Ad}}_{T}=\left[\begin{array}{cc} R & 0\\ {}[t]R & R \end{array}\right],$$
where R is the rotational part of T, and [t] denotes the skew-symmetric matrix corresponding to the translation vector t.

When a link is subjected to a force f and a torque n, the wrench is defined for power computation purposes as $F={\left[\begin{array}{cc} n^{T} & f^{T} \end{array}\right]}^{T}$. The transformation of the wrench from frame i to frame i−1 is expressed as:(3)$$F_{i-1}={\left({\mathrm{Ad}}_{T_{i-1,i}^{-1}}\right)}^{T}F_{i},$$
where $F_{i}$ and $F_{i-1}$ are the wrenches expressed in link frames i and i−1, respectively; ${\left({\mathrm{Ad}}_{T_{i-1,i}^{-1}}\right)}^{T}$ denotes the transpose of the adjoint transformation matrix corresponding to the inverse transformation $T_{i-1,i}$.

The above-defined twists and wrenches are vector representations of elements in the Lie algebra se(3). Twists can also undergo the Lie bracket operation, which can be interpreted as a generalized cross product in the six-dimensional space:(4)V2=adV0V1=[ω0]0[v0][ω0]V1,
where V0, V1, and V2 are all velocity twists; adV0∈ℝ6×6 is the matrix representation of the Lie bracket operator; [ω0] and [v0] denote the skew-symmetric matrices corresponding to the angular velocity ω0 and linear velocity v0, respectively.

### 2.2. Dynamic Modeling of Robots Using the Newton–Euler Method

The Newton–Euler method involves two recursive steps: first, propagating the velocities and accelerations from the robot base to the end-effector; second, computing the forces and torques in reverse from the end-effector back to the base. In both steps, link-fixed coordinate frames must be defined to ensure consistent interpretation of physical quantities.

Most studies [10,11,22,23] begin by explicitly defining the homogeneous transformation between adjacent link frames. However, when using the product-of-exponentials (PoE) formula to formulate the kinematic model, such transformations are not required. Instead, we adopt the same coordinate frame convention used in the PoE-based kinematic modeling, where each link frame is naturally determined by the joint screw axes and the origin vectors defined in the fixed base frame. This approach ensures consistency with the geometric structure of the PoE formulation and avoids the need for additional frame assignments.

As shown in Figure 2, the *z*-axis of the link-fixed frame, expressed in the fixed frame, is denoted by vector *s*. To align this axis with the fixed frame’s *z*-axis, denoted by $z={\left[\begin{array}{ccc} 0 & 0 & 1 \end{array}\right]}^{T}$, a rotation about the screw axis l=s×z by an angle $\theta =\arccos\left(\frac{s\cdot z}{\left|s||z\right|}\right)$ is required. Using Rodrigues’ formula, the rotation matrix that maps the link-fixed frame to the fixed frame is given by:(5)R=I+[l]sinθ+[l][l](1−cosθ),

The translational component of the homogeneous transformation is simply the origin vector *p*. The corresponding homogeneous transformation matrices mapping each link-fixed frame to the fixed frame are denoted as $M_{0,i}$. The homogeneous transformation matrix between adjacent link-fixed frames is then given by:(6)$$M_{i-1,i}={\left(M_{0,i-1}\right)}^{-1}M_{0,i}.$$

The homogeneous transformation matrix between adjacent link-fixed frames after robot motion becomes:(7)Ti−1,i=Mi−1,ie[ξi]θi,
where Ti−1,i denotes the homogeneous transformation matrix between adjacent link-fixed frames at an arbitrary time instant; e[ξi]θi represents the matrix exponential of the Lie algebra element [ξi]θi, where θi is the joint rotation angle, and [ξi] is the matrix representation of the screw coordinate [ξi]=[z]z000∈ℝ4×4. In the *i*-th link-fixed frame, the *z*-axis is axis of rotation, ξi=zTz0TT=[0, 0, 1, 0, 0, 0]T.

With the link-fixed coordinate frames defined, the velocity and acceleration screws of each link can be computed recursively.(8)$$\left\{\begin{array}{l} V_{i}={\mathrm{Ad}}_{T_{i-1,i}^{-1}}V_{i-1}+{\dot{\theta }}_{i}\xi_{i}\\ {\dot{V}}_{i}={\mathrm{Ad}}_{T_{i-1,i}^{-1}}{\dot{V}}_{i-1}-{\mathrm{ad}}_{{\dot{\theta }}_{i}\xi_{i}}V_{i}+{\ddot{\theta }}_{i}\xi_{i} \end{array}\right.,$$
where $V_{i-1}$ and $V_{i}$ denote the spatial velocity twists of link i−1 and link i, respectively, expressed in their respective link-fixed coordinate frames. The term ${\dot{\theta }}_{i}\xi_{i}$ represents the velocity twist generated by the joint *i*.

The force analysis of each link starts from the end-effector and proceeds backward to the base. The force of link and joint torque are:(9)Fi=GiV˙i−adViTGiVi+AdTi,i+1−1TFi+1τi=FiTξi,
where Gi∈ℝ6×6 is the spatial inertia matrix of link *i* expressed in its own coordinate frame:(10)Gi=Iimi[rci]−mi[rci]miI3×3,

Here, Ii∈ℝ3×3 is the inertia tensor of link *i*, mi is its mass, rci∈ℝ3 is the vector from the origin to the center of mass in the link-fixed frame, and I3×3 is the 3 × 3 identity matrix.

Since we have defined the link-fixed coordinate frames in advance, the spatial inertia matrix of each link remains constant in its own frame. For link *i*, there are 10 independent inertial parameters in its own coordinate frame: six related to the inertia tensor—Ixxi, Iyyi, Izzi, Ixyi, Iyzi, Ixzi, three related to the center of mass—mircxi, mircyi, mirczi, and one for the total mass mi. These are collected into a parameter vector λi∈ℝ10×1. Using the vectorization operator, the columns of the spatial inertia matrix Gi are stacked into a single vector, denoted as vec(Gi)∈ℝ36×1. There exists a matrix K∈ℝ36×10 composed solely of 0 s, 1 s, and −1 s such that:(11)vec(Gi)=Kλi.

## 3. Minimal Parameter Set Extraction and Forward Dynamics

### 3.1. Block Matrix Representation of Recursive Dynamics

For the entire robotic system, the recursive formulations reflect the dynamic coupling between adjacent links. This recursive relationship can be expressed in a block matrix form, which facilitates both computational implementation and further analytical treatment; **see details in**
**[8,10]**.(12)V=ΓAθ˙+W0V0V˙=Γ−[adθ˙ξ]V+Aθ¨+W0V˙0,
where V0=[0, 0, 0, 0, 0, 0]T is the twist of the robot base. V˙0∈ℝ6×1 denotes the acceleration screw of the robot base. Since the base frame is fixed in the inertial frame and gravity acts along the negative *z*-axis, we have V˙0=[0, 0, 0, 0, 0, −9.8]T. Detailed symbol representations can be found in Appendix A.

The matrix equation for the overall wrench vector **F** and joint torques becomes:(13)F=ΓT−[adV]TGV+GV˙+Wn+1Fn+1τ=ATF,
where τ=[τ1, τ2, …, τn]T∈ℝn×1.

Substituting Equations (12) and (13) into each other yields the final expression for the joint torque vector:(14)$$\tau =A^{T}\Gamma^{T}G\Gamma A\ddot{\theta }-A^{T}\Gamma^{T}{\left[{\mathrm{ad}}_{V}\right]}^{T}G\Gamma A\dot{\theta }-A^{T}\Gamma^{T}G\Gamma [{\mathrm{ad}}_{\dot{\theta }\xi }]\Gamma A\dot{\theta }+A^{T}\Gamma^{T}G\Gamma W_{0}{\dot{V}}_{0}+A^{T}\Gamma^{T}W_{n+1}F_{n+1}.$$

### 3.2. Equivalent Transformation of Linear Matrix Equations and Calculation of the Minimal Inertial Parameter Set

When the robot is not subjected to any external wrench $F_{n+1}$, Equation (14) can be simplified and combined into:(15)$$\tau =\left(A^{T}\Gamma^{T}\right)G\left(\Gamma A\ddot{\theta }-\Gamma \left[{\mathrm{ad}}_{\dot{\theta }\xi }\right]\Gamma A\dot{\theta }+\Gamma W_{0}{\dot{V}}_{0}\right)+\left(-A^{T}\Gamma^{T}{\left[{\mathrm{ad}}_{V}\right]}^{T}\right)G\left(\Gamma A\dot{\theta }\right),$$

Let us define the following matrices:(16)A=ATΓT∈ℝn×6n,B=ΓAθ¨−Γadθ˙ξΓAθ˙+ΓW0V˙0∈ℝ6n×1,C=−ATΓTadVT∈ℝn×6n,D=ΓAθ˙∈ℝ6n×1.

The joint torque vector of the entire robotic system is then expressed as:(17)τ=AGB+CGD.

In numerical analysis, linear matrix equations of the form AXB+CXD=E can be equivalently transformed into a vector–matrix equation using the Kronecker product and the vectorization operator [10,12,13,24]:(18)$$\left(B^{T}⊗A+D^{T}⊗C\right)\mathrm{vec}(G)=\tau .$$

However, in practice, joint friction significantly affects the accuracy of dynamic modeling. Although various friction models have been proposed for robotic systems, the most commonly used are the viscous and Coulomb friction components [22,25]. Therefore, the dynamic model is modified to include these effects:(19)$$\left(B^{T}⊗A+D^{T}⊗C\right)\mathrm{vec}(G)+f_{v}\dot{\theta }+f_{c}\mathrm{sgn}(\dot{\theta })=\tau ,$$
where $f_{v}$ and $f_{c}$ denote the diagonal matrices of viscous and Coulomb friction coefficients, respectively.

Since G is a block-diagonal matrix, we can simplify the above formulation by following the method outlined in [10]. Partitioning the coefficient matrices accordingly, let Aij∈ℝ1×6, Bj∈ℝ6×1, Cij∈ℝ1×6, Dj∈ℝ6×1, and define:(20)yij=BjT⊗Aij+DjT⊗Cij∈ℝ1×36.

Combining this with Equation (19), we obtain:(21)$$Y_{1}\lambda_{1}+Y_{2}\lambda_{2}=Y\lambda =\tau ,$$
where:

Y1=y11y12…y1ny21y22…y2n⋮⋮⋱⋮yn1yn2…ynnK0…00K…0⋮⋮⋱⋮00…K∈ℝn×10n, *K* from Equation (11).

Y2=θ˙1sgn(θ˙1)00…0000θ˙2sgn(θ˙2)…00⋮⋮⋮⋮⋱⋮⋮0000…θ˙nsgn(θ˙n)∈ℝn×2n,



$Y=[Y_{1}\;Y_{2}],$



λ1=[λ1T λ2T … λnT]T∈ℝ10n×1,



λ2=[fv1 fc1 fv2 fc2 … fvn fcn]T∈ℝ2n×1,



$\lambda =\left[\begin{array}{c} \lambda_{1}\\ \lambda_{2} \end{array}\right].$



The coefficient matrix Y depends on the robot’s joint positions, velocities, and accelerations. For different robot configurations, both Y and τ vary. A necessary condition for the vector–matrix equation to admit a unique solution is that Y must be of full column rank. Therefore, at least 12 sets of sampling data are required to potentially achieve full column rank.

It is worth noting that the friction-related submatrix $Y_{2}$ is always of full column rank, while $Y_{1}$ may contain linearly dependent or zero columns. To address this, a full-rank decomposition is applied to $Y_{1}$.

By collecting m sets of sampling data, the matrix $Y_{1}$ is expanded into:(22)H1=Y1T1Y1T2…Y1TmT∈ℝmn×10n.

The full-rank decomposition is performed as follows:

1.Apply Gaussian elimination to transform $H_{1}$ into reduced row echelon form. The column indices corresponding to the pivot elements represent the linearly independent columns of $H_{1}$.2.The non-zero rows of the resulting matrix form the right factor of the decomposition, while the corresponding independent column vectors from the original matrix constitute the left factor.

This yields:(23)H1=H0K,
where H0∈ℝmn×r is of full column rank, and K∈ℝr×10n is of full row rank.

Using this decomposition, Equation (21) can be rewritten as:(24)$$\left[\begin{array}{cc} H_{0} & H_{2} \end{array}\right]\left[\begin{array}{c} \lambda_{0}\\ \lambda_{2} \end{array}\right]=H\lambda_{m}=\tau ,$$
where:

H2=Y2T1Y2T2…Y2TmT∈ℝmn×2n,τ=τT1τT2…τTmT∈ℝmn×1,λm∈ℝ(r+2n)×1 is the minimal parameter set,λ0=Kλ1∈ℝr×1 is the reduced-order parameter vector.

All the above transformations are linear. For any sampled $H_{1}$, the full-rank decomposition yields the same full-row-rank matrix K. Hence, the identified minimal parameter set $\lambda_{m}$ is uniquely determined. To determine whether Equation (24) admits a unique solution, we compute the rank of the augmented matrix [H τ]. If the rank is equal to r+2n, a unique solution exists. However, if the rank equals r+2n+1, it indicates significant measurement noise or error in the sampled joint torques, making parameter estimation unreliable.

### 3.3. Forward Dynamics

The identified dynamic parameters are only meaningful when they can be consistently applied in both forward and inverse dynamics. While the recursive Newton–Euler formulation inherently provides an inverse dynamic model, forward dynamics require the computation of the mass matrix M(θ). From Equation (14), the mass matrix can be expressed as:(25)$$\left\{\begin{array}{l} M(\theta )=A^{T}\Gamma^{T}\mathrm{G\Gamma A}\\ \mathrm{vec}(M)=(A⊗A)\mathrm{vec}\left(G\right) \end{array},\right.$$
where the linearization technique for matrix equations of the form AXB=C is applied.

To incorporate the minimal inertial parameter set obtained earlier, this expression can be further simplified as:(26)$$\mathrm{vec}(M)=ZK\lambda_{1}=Z_{0}\lambda_{0},$$
where:

vec(M)∈ℝn2×1 is the vectorized form of the inertia matrix Mθ;K is a block-diagonal extension of the matrix K defined in Equation (11);Z0 consists of selected columns from ZK corresponding to the same column indices used in the extraction of H0.

Partitioning the matrix A into blocks such that each sub-block Aij∈ℝ1×6, we define the elements of Z as:(27)Zmj=Aij⊗Azj∈ℝ1×36, m=n·i+z−n,
where i,j,z∈{1,…,n}.

Given the reduced-order parameter vector $\lambda_{0}$, the vector vec(M) can be computed using Equation (26), from which the full inertia matrix M(θ) is reconstructed.

With the inertia matrix available, the forward dynamic equation is given by:(28)$$\ddot{\theta }=M^{-1}(\theta )(\tau -Y_{0}(\theta ,\dot{\theta },0)\lambda^{0}-Y_{2}(\dot{\theta })\lambda^{2}),$$
where:

$Y_{0}(\theta ,\dot{\theta },0)$ denotes the regressor matrix $Y_{0}$ evaluated with zero joint accelerations, but known joint positions and velocities;$Y_{2}(\dot{\theta })$ is the friction-related regressor matrix;$\lambda_{0}$ and $\lambda_{2}$ are the reduced-order inertial and frictional parameter vectors, respectively.

This formulation enables real-time simulation and control of the robot system based on the identified minimal parameter set.

## 4. Excitation Strategy

The vector–matrix equation given in Equation (24) can be solved using the least squares method, where the estimation error mainly arises from two sources: the excitation trajectory and the measurement noise in the joint torques. The excitation trajectory directly affects the coefficient matrix H, and its condition number determines the sensitivity of the solution to measurement errors [26]. In general, a larger condition number results in a greater estimation error. This relationship can be bounded as:(29)$$\frac{\left\|\Delta \lambda_{m}\right\|}{\left\|\lambda_{m}\right\|}\leq \mathrm{cond}(H)\cdot \frac{\left\|\Delta \tau \right\|}{\left\|\tau \right\|},$$
where:

cond(H) denotes the condition number of the coefficient matrix H;$\Delta \lambda_{m}$ is the parameter estimation error vector;Δτ is the joint torque measurement error vector.

Minimizing the condition number of H can therefore serve as an objective function for optimizing the excitation trajectory. Although finite Fourier series can provide flexibility in trajectory design, solving for the optimal trajectory is computationally intensive and may only yield a local optimum.

An alternative approach is to perform single-joint excitation, which results in a sparse structure in the coefficient matrix H. This structure allows further reduction of the parameter vector through full-rank decomposition and facilitates minimization of the condition number. Using sinusoidal functions for single-joint excitation ensures smooth start-up and non-impulsive motion. Moreover, increasing the frequency of the sine wave generally reduces the condition number.

The excitation trajectory for each joint is defined as:(30)$$\theta_{i}=a_{i}\sin(w_{i}\cdot t),$$
where, $\theta_{i}$ is the angular position of joint i; $a_{i}$ is the amplitude of the sine wave; $w_{i}$ is the angular frequency.

In this work, the following criteria are applied when designing the excitation strategy for dynamic parameter identification:

1.Sequential Joint Excitation: Start from the distal joints and excite one joint at a time while keeping all other joints fixed at safe positions.2.Amplitude determination: The amplitude $a_{i}$ determines the range of motion for each joint and is selected based on the physical constraints of the corresponding joint.3.Frequency tuning: To ensure robot safety, the frequency $w_{i}$ is gradually increased until the maximum joint velocity remains within the allowable limits. A relatively high frequency is preferred to reduce the condition number.

During single-joint excitation, although only one joint is actively driven, the passive joints still experience torque variations due to dynamic coupling effects. Therefore, joint torques across all joints must be sampled simultaneously.

Joint position data are typically obtained from motor encoders. Joint velocities and accelerations are estimated through first- and second-order numerical differentiation of the encoder signals. In parameter identification, the actual sampled signals—rather than the planned trajectory values—are used to construct the regressor matrix H. This is because most robotic controllers employ PID (Proportional–Integral–Derivative) control, leading to phase lag and tracking errors between the reference and actual joint positions. Using measured joint states together with sampled torque data helps maintain consistency in the identification process. However, such sampling-induced errors inevitably increase the discrepancy between the identified parameters and their true values.

## 5. Parameter Identification and Experimental Validation

The experiments were conducted using a custom-designed horizontal lower-limb rehabilitation robot developed in our laboratory. The first three joints of the robot form a planar 3R mechanism, while the fourth joint operates in a plane perpendicular to that of the first three joints. Furthermore, the end-effector has relatively low mass, and the fourth joint exhibits negligible dynamic coupling with the first three joints. Therefore, for the purpose of dynamic modeling and parameter identification, the system can be effectively treated as a planar 3R manipulator.

As shown in Figure 3, the robotic system consists of an industrial personal computer, a controller, servo drivers, servo motors, and the robot body. After designing the excitation trajectories, motion programs are implemented, and feedback data are processed via the upper-level computer. The real-time controller (LinksTech-II) executes the control algorithms and records essential data during operation.

Each axis motor is driven by a Yaskawa single-axis servo driver, which follows pulse commands from the controller. The servo drivers also monitor the motor currents and output estimated motor torque as analog voltage signals via the CN5 interface. The multifunction I/O board PCI6259 in the controller acquires the joint torque data from these analog signals.

The motor encoder provides real-time angular position feedback. In the controller, the actual joint angles are computed based on the known gear transmission ratios. Similarly, the motor torques are converted into joint torques using the same transmission ratios. After completing the robot motion, the recorded joint angle and torque data are exported from the controller for dynamic parameter identification.

### 5.1. Minimal Parameter Set Identification and Excitation Trajectory Analysis

Establish the link coordinate frames at the joints of the robot used in the experiment, as shown in Figure 4. The origin of the base frame (Frame 0) coincides with that of Link 1, and its z-axis is oriented vertically upward. The x-axis of the base frame points in the direction opposite to the z-axis of Frame 1.

Based on this configuration, the joint axis direction vectors $s_{i}$ and the position vectors of the frame origins $p_{i}$, both expressed in the base frame, are obtained as listed in Table 1.

Using the formulation derived in Section 2 and Section 3, a symbolic computation program was implemented in MATLAB (2023a) to automatically calculate the coefficient matrix Y in Equation (24). A set of randomly generated joint positions θ, velocities $\dot{\theta }$, and accelerations $\ddot{\theta }$ were used as input data to construct the extended regressor matrix $H_{1}$ from the original symbolic matrix $Y_{1}$.

The MATLAB function `rref` was employed to compute the reduced row echelon form of $H_{1}$. This operation yielded a full-row-rank matrix K, where the rank was equal to the number of non-zero rows. The column indices corresponding to the pivot columns of the reduced matrix indicate the linearly independent columns of $H_{1}$.

For the robot under study, 12 sets of random input data were used. Through repeated computations, it was observed that regardless of the input data variation, the resulting reduced row echelon forms and the linearly independent column indices remained consistent. This consistency confirms the structural uniqueness of the minimal parameter set for this robotic system.

The matrix formed by selecting these linearly independent columns, denoted $Z_{0}$, was then used to extract the mass matrix M(θ), which plays a central role in forward dynamics modeling.

The amplitude of the excitation trajectory for each joint was determined based on its allowable motion range, as summarized in Table 2. The frequency values were gradually increased during experiments to ensure safety while minimizing the condition number of the regressor matrix H. As expected, increasing the excitation frequency generally led to a reduction in the condition number.

The excitation was applied sequentially starting from the third joint. The resulting condition numbers of H were 47.73, 52.95, and 54.34 for joints 3, 2, and 1, respectively.

Prior to parameter identification, the linearly independent column indices were used to extract the corresponding columns from the symbolic matrix $Y_{1}$, forming a full-column-rank symbolic matrix $Y_{0}$. This matrix, together with the friction-related regressor matrix $Y_{2}$, was saved for use in subsequent identification tasks.

The inertial parameters $\lambda_{0}$ and the joint friction parameters $\lambda_{2}$ were combined into a single minimal parameter vector $\lambda_{m}$, which fully characterizes the dynamic behavior of the robot. The identified minimal parameter set is presented in Table 3.

The controller operated at a sampling frequency of 1000 Hz. During each excitation trajectory, partial torque values for each joint were recorded, as shown in Figure 5, Figure 6 and Figure 7. Since the excitation trajectories were periodic, only one cycle of data was used for analysis. Although the encoder readings for joint angles were relatively noise-free, the angular velocities and accelerations—obtained via numerical differentiation—introduced significant noise. To mitigate this, the Zero Phase Filtering method was applied to the velocity and acceleration signals, preserving signal integrity without introducing phase lag.

Filtered joint angle, velocity, and acceleration data were substituted into the precomputed symbolic matrices $Y_{0}$ and $Y_{2}$. Using the least-squares method on the regression equation (Equation (24)), the unknown parameter vector $\lambda_{m}$ was estimated. Given that only one joint was actively excited at a time, the identification results primarily reflected the inertial parameters of the moving joint. Therefore, only the relevant subset of parameters was selected for updating in each trial. The complete minimal parameter set obtained through this process is summarized in Table 3.

### 5.2. Experimental Validation and Analysis

To validate the accuracy of the identified parameters, Fourier series were utilized to generate smooth, periodic joint trajectories. These trajectories were executed on the physical robot while recording motor current-based torque estimates and simultaneously fed into the inverse dynamics model using the identified minimal parameter set $\lambda_{m}$ to compute the expected joint torques. Figure 8, Figure 9 and Figure 10 illustrate the comparison between the measured torque values and the estimated torque values derived from the inverse dynamics calculations using the identified parameters.

Key observations:

Consistency in trend: The estimated torque values closely follow the trend of the measured torque values, indicating that the identified parameters accurately capture the dynamic characteristics of the robot.Reflection of torque variations: The estimated torques not only match the overall trend but also reflect detailed variations observed in the experimental data, demonstrating the effectiveness of the parameter identification process.

Figure 11 illustrates the error distribution between the estimated and filtered measured joint torques. The root mean square errors (RMSEs) for joints 1, 2, and 3 are 23.76 N·m, 17.09 N·m, and 3.55 N·m, respectively. For joint 1, 50% of the errors fall within −18.73 N·m to 11.17 N·m, with maximum and minimum errors of 50.3 N·m and −86.59 N·m, respectively, and there are 42 outliers. For Joint 2, 50% of the errors fall within −17.72 N·m to 4.83 N·m, with maximum and minimum errors of 36.37 N·m and −45.2 N·m, respectively. For Joint 3, 50% of the errors fall within −1.18 N·m to 3.88 N·m, with maximum and minimum errors of 7.6 N·m and −7.86 N·m, respectively.

The primary sources of error include:

Inaccuracies in model identification: Factors such as installation errors and poor lubrication of gearboxes contribute to large friction parameter values.Measurement accuracy: Joint torque measurements, derived from motor current measurements by the actuator, introduce measurement noise and motor friction torque, leading to significant errors at the joint level.

The minimal parameter set was further applied to forward dynamics, and joint accelerations were computed using Equation (28), as shown in Figure 12, Figure 13 and Figure 14. While there were discrepancies between the estimated and measured joint accelerations, their temporal trends are in good agreement. The estimation errors in the minimal parameter set clearly influenced the forward dynamics estimates, the consistency in trends suggests that the minimal parameter set can be effectively used for forward dynamics predictions.

## 6. Conclusions

Screw theory provides a holistic framework for describing the velocity and force transformations between robotic links, enabling a compact and coordinate-invariant formulation of the Newton–Euler dynamic equations [8]. Moreover, by employing the Kronecker product to transform the linear matrix equation into an equivalent vector–matrix form, the complexity of solving the dynamic model is significantly reduced. This transformation facilitates the identification of inertial parameters in robot dynamics and makes the overall process more computationally efficient [10]. However, only the minimal parameter set, which consists of a unique and identifiable subset of all inertial and friction parameters, can be practically estimated from experimental data. Utilizing this minimal set ensures that the resulting identification is both meaningful and robust for real-world applications.

Main contributions:

1.Efficient extraction of the unique minimal parameter set via full-rank decomposition. Once the link coordinate frames are defined, the linearized dynamic model yields a regressor matrix Y. Despite being time-varying, this matrix exhibits an invariant set of pivot columns across all motion states. By performing full-rank decomposition on an extended version of the regressor matrix (denoted H), we consistently obtain the same full row-rank matrix and the same set of linearly independent column indices, regardless of the input trajectory. This property guarantees the uniqueness and repeatability of the minimal parameter set extraction process.2.Minimal parameter set application to forward dynamics modeling. The identified minimal parameter set is not only suitable for inverse dynamics estimation but also enables forward dynamics modeling. Specifically, the mass matrix—central to forward dynamics—can be reconstructed using the Kronecker product-based vectorization technique combined with the minimal parameter set. This allows us to derive the forward dynamics model directly from the identified inverse dynamics model, providing a solid foundation for simulation, control design, and real-time prediction of robotic motion behavior.3.Sequential joint excitation strategy for safe and accurate identification. In our identification procedure, only one joint is actuated at a time, resulting in a single-input multiple-output (SIMO) identification scheme. This sequential excitation strategy ensures operational safety while allowing for targeted identification of joint-specific inertial parameters. Additionally, by tuning the excitation frequency, the condition number of the regressor matrix can be effectively reduced, thereby improving the numerical stability and accuracy of the least-squares estimation.

The proposed method was validated on a planar 3R serial robot. Experimental results confirm that the sequential joint excitation strategy enables safe and accurate identification of the minimal parameter set, and the reconstructed forward dynamics model closely replicates the actual robot’s dynamic response. Although the current study focuses on a low-degree-of-freedom robot system, the methodology naturally extends to 6-DOF industrial manipulators. The core principles—coordinate frame assignment, symbolic regressor construction, full-rank decomposition, and sequential excitation—remain valid regardless of the robot’s complexity.

## Figures and Tables

**Figure 1 sensors-25-05749-f001:**
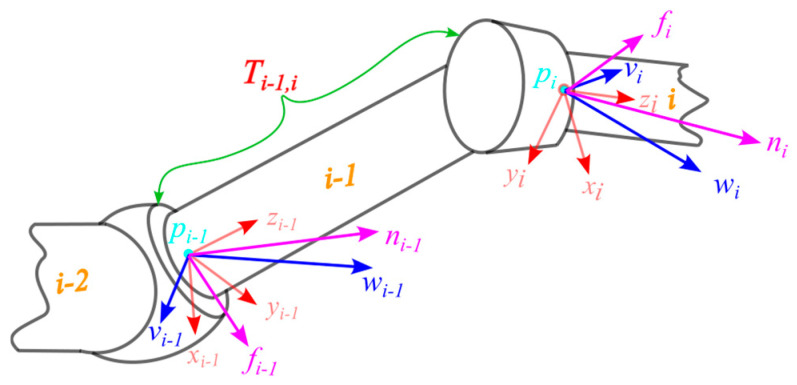
Homogeneous transformation of velocity twists and wrenches.

**Figure 2 sensors-25-05749-f002:**
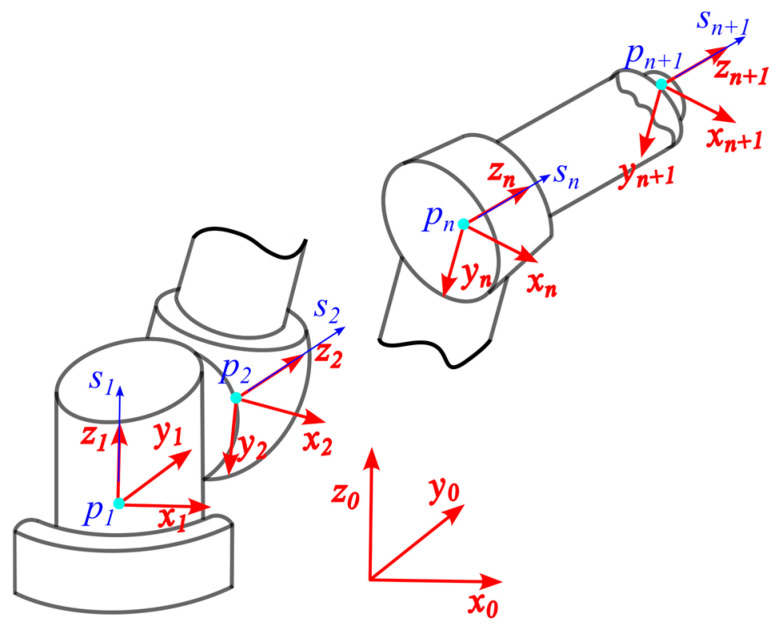
Definition of link frames and fixed frame.

**Figure 3 sensors-25-05749-f003:**
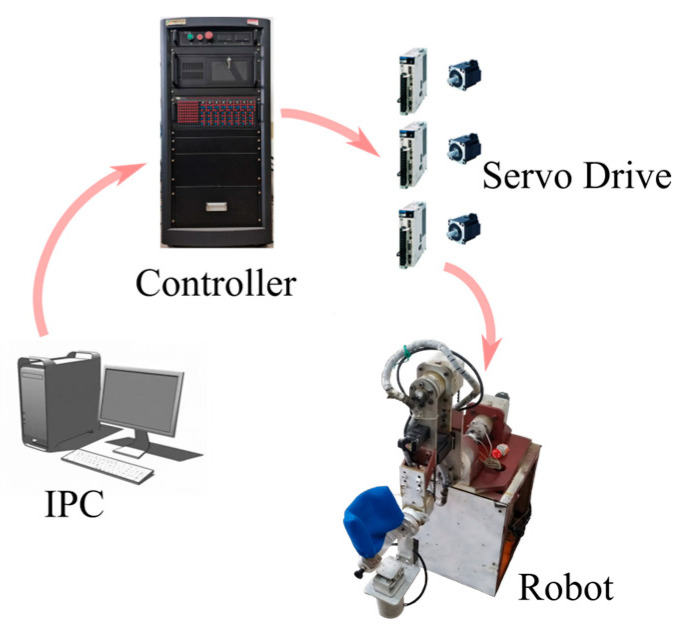
Horizontal lower limb rehabilitation robot system.

**Figure 4 sensors-25-05749-f004:**
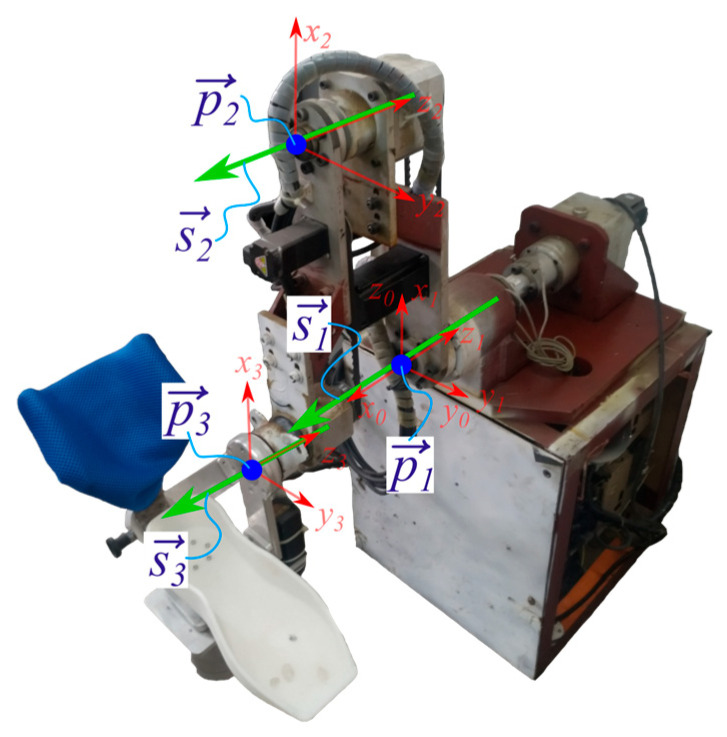
Link frame of robot.

**Figure 5 sensors-25-05749-f005:**
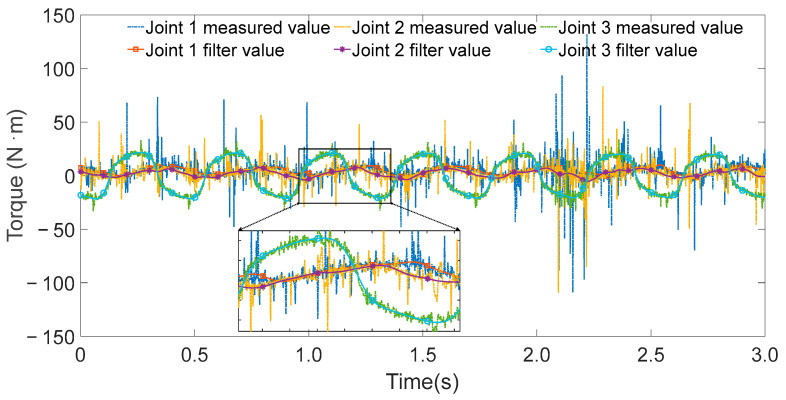
All joint torque values and filtering values when exciting joint 3.

**Figure 6 sensors-25-05749-f006:**
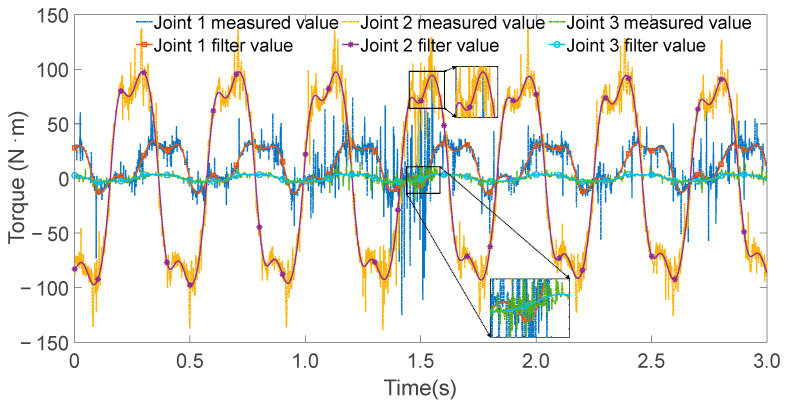
All joint torque values and filtering values when exciting joint 2.

**Figure 7 sensors-25-05749-f007:**
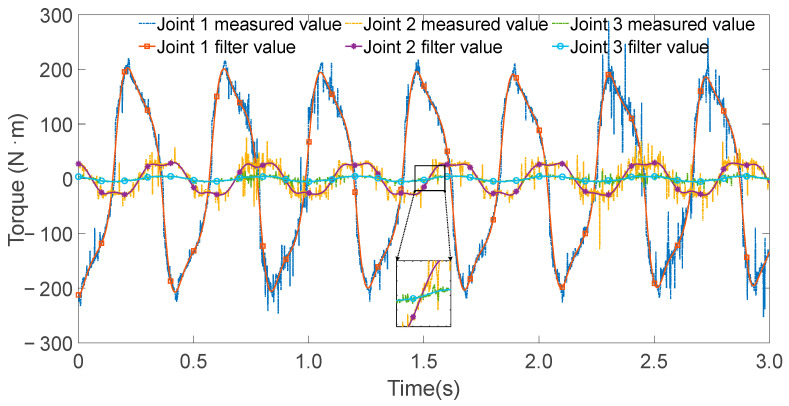
All joint torque values and filtering values when exciting joint 1.

**Figure 8 sensors-25-05749-f008:**
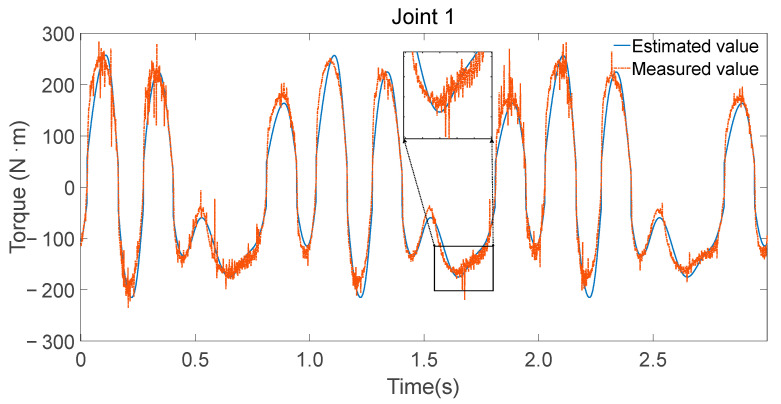
Measurement and estimation of joint 1 torque.

**Figure 9 sensors-25-05749-f009:**
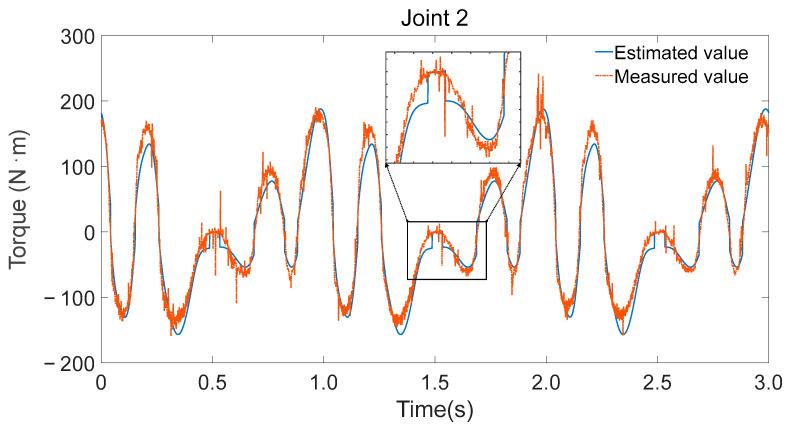
Measurement and estimation of joint 2 torque.

**Figure 10 sensors-25-05749-f010:**
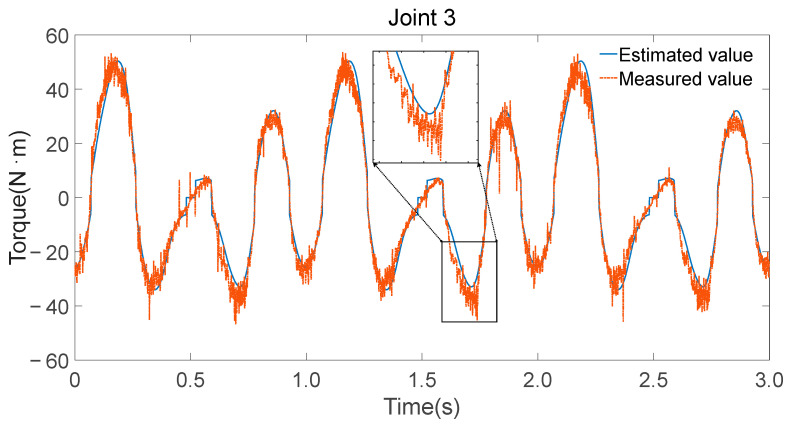
Measurement and estimation of joint 3 torque.

**Figure 11 sensors-25-05749-f011:**
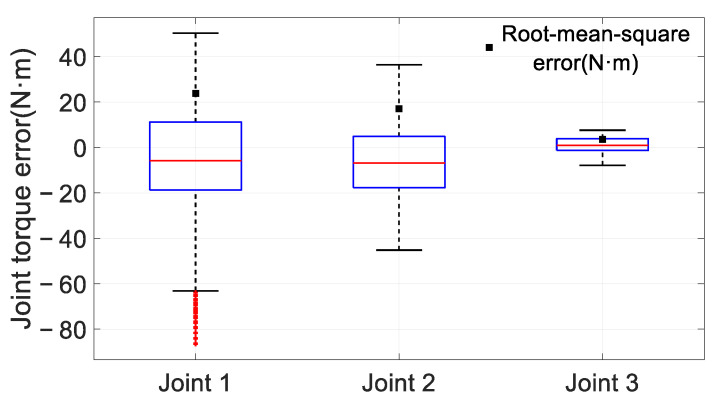
Statistical analysis of the error between the filtered and estimated values of joint torque.

**Figure 12 sensors-25-05749-f012:**
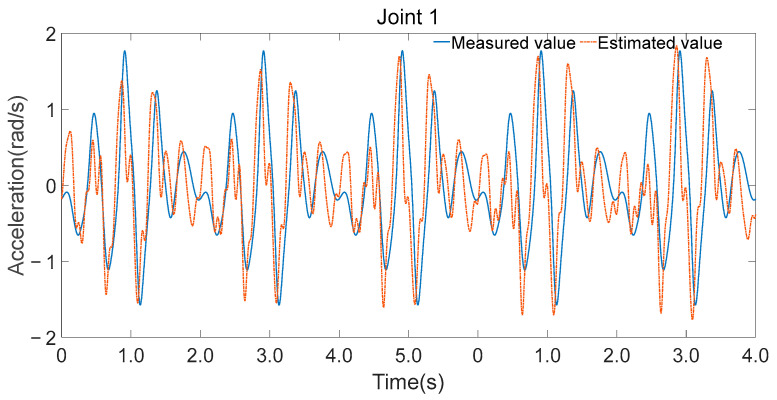
Estimated and measured angular acceleration of joint 1.

**Figure 13 sensors-25-05749-f013:**
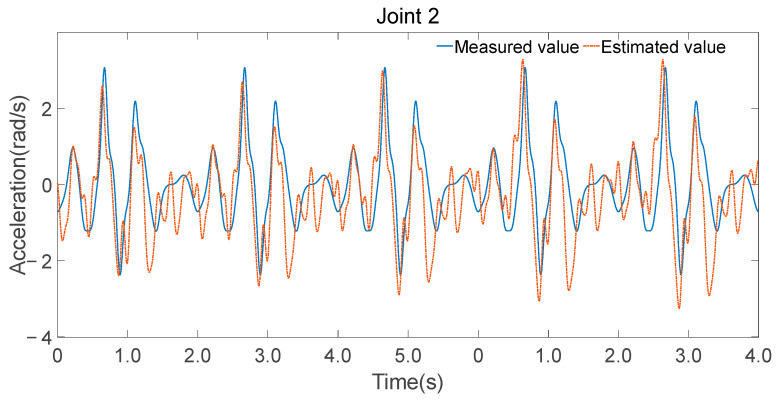
Estimated and measured angular acceleration of joint 2.

**Figure 14 sensors-25-05749-f014:**
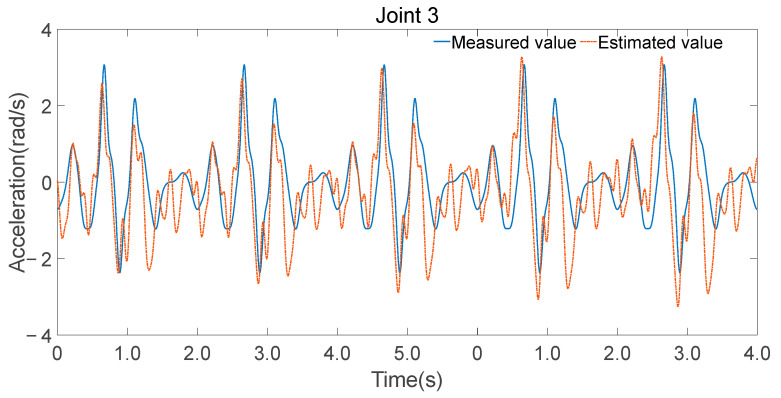
Estimated and measured angular acceleration of joint 3.

**Table 1 sensors-25-05749-t001:** Axis Direction Vectors and Frame Origin Vectors.

Frame Index	$\mathrm{Axis}\;\mathrm{Direction}\;\mathrm{Vector}\;s_{i}$/m	$\mathrm{Frame}\;\mathrm{Origin}\;\mathrm{Vector}\;p_{i}$/m
1	$s_{1}={\left[\begin{array}{ccc} 1 & 0 & 0 \end{array}\right]}^{T}$	$p_{1}={\left[\begin{array}{ccc} 0 & 0 & 0 \end{array}\right]}^{T}$
2	$s_{2}={\left[\begin{array}{ccc} 1 & 0 & 0 \end{array}\right]}^{T}$	$p_{2}={\left[\begin{array}{ccc} 0.12 & 0 & 0.36 \end{array}\right]}^{T}$
3	$s_{3}={\left[\begin{array}{ccc} 1 & 0 & 0 \end{array}\right]}^{T}$	$p_{2}={\left[\begin{array}{ccc} 0.2425 & 0 & 0 \end{array}\right]}^{T}$

**Table 2 sensors-25-05749-t002:** Amplitude $a_{i}$ and Frequency $w_{i}$ of Excitation Trajectories.

Joint Index	$\mathrm{Amplitude}\;a_{i}$/rad	$\mathrm{Frequency}\;w_{i}$/rad/s
1	5π/9	15
2	2π/3	15
3	π/2	14

**Table 3 sensors-25-05749-t003:** Identified Minimal Parameter Set $\lambda_{m}$.

Symbol	Value (SI Units)	Symbol	Value (SI Units)
${𝕀}_{zz1}+0.1296m_{2}+0.1296m_{3}$	5.888	$m_{3}r_{cy3}$	0.0016
$m_{1}r_{cx1}+0.36m_{2}+0.36m_{3}$	5.199	$f_{v1}$	−157.56
$m_{1}r_{cy1}$	0.0469	$f_{c1}$	−40.402
$m_{1}r_{cy1}$	3.588	$f_{v2}$	−46.302
$m_{2}r_{cx2}-0.36m_{3}$	−1.284	$f_{c2}$	−12.837
$m_{2}r_{cy2}$	0.0381	$f_{v3}$	−9.562
${𝕀}_{zz3}$	0.111	$f_{c3}$	−2.922
$m_{3}r_{cx3}$	−0.287		

## Data Availability

Data can be accepted on request.

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
