# Peer review of "Research on Identification of Minimum Parameter Set in Robot Dynamics and Excitation Strategy"

_sensors, 2025, doi:10.3390/s25185749_

Round 1
Reviewer 1 Report
Comments and Suggestions for Authors
The work has merit and potential for publication; however, it requires important revisions before it can be accepted, which are listed below.
1 - The abstract and introduction state that the work addresses the complexity and limited applicability of existing methods for determining the minimum set of dynamic parameters, mentioning that many are complex and computationally intensive, with non-intuitive solving procedures. However, the article lacks a more in-depth and quantitative comparison with these specific methods. It would be beneficial to include a more detailed discussion, possibly with a comparative table, that quantifies the advantages of the proposed approach in terms of computational time, accuracy, or robustness compared to the cited methods (e.g., [2–4, 17–21]). This would strengthen the claim of significant reduction in complexity.
2- The Root Mean Square Error (RMSE) values for the joint torques are provided: 23.76 N·m for joint 1, 17.09 N·m for joint 2, and 3.55 N·m for joint 3. Especially for joints 1 and 2, these values seem considerably high for an application aiming for precise control and faithful replication of the robot's kinematic behavior.
3- The error ranges and maximum/minimum values (e.g., for Joint 1: maximums of 50.3 N·m and minimums of -86.59 N·m), as well as the presence of 42 outliers for Joint 1, are concerning. The authors attribute the sources of error to inaccuracies in model identification (installation errors, poor gearbox lubrication leading to large friction parameter values) and measurement accuracy (noise, motor friction torque. While these are plausible explanations, I suggest that the authors:
• Discuss more deeply the practical implications of these errors on the real-world application of the model, especially in scenarios requiring high-precision control or realistic simulations (e.g., for RL training). • If possible, quantify the impact of the identified large friction parameter values (Table 3) on the overall model accuracy. 4- The graphs presented in Figures 5 through 10 should be redone, as their current quality is inadequate. In Figures 5, 6, and 7, it is particularly difficult to discern the filtered signal lines, which impedes a clear and accurate analysis of the results. 5- The article incorporates viscous and Coulomb friction components, which is a common practice. However, a brief justification for choosing this specific friction model over other more complex models (such as the Stribeck model) could enrich the discussion, especially given that the friction parameters were identified and contributed to the observed errors. The manuscript offers a valuable contribution to the identification of robot dynamic parameters and their application in forward and inverse dynamic models, a topic of great relevance to the field of sensors and robotics. The methodology is well-articulated, and the experimental validation is a strong point. The suggested areas for improvement aim to deepen the comparative analysis and detail the implications of the observed errors, which would help to position the work more robustly in the context of the precision required for advanced robotics applications. After the incorporation of these revisions, i recomend the publication.
Author Response
Comments 1: The abstract and introduction state that the work addresses the complexity and limited applicability of existing methods for determining the minimum set of dynamic parameters, mentioning that many are complex and computationally intensive, with non-intuitive solving procedures. However, the article lacks a more in-depth and quantitative comparison with these specific methods. It would be beneficial to include a more detailed discussion, possibly with a comparative table, that quantifies the advantages of the proposed approach in terms of computational time, accuracy, or robustness compared to the cited methods (e.g., [2–4, 17–21]). This would strengthen the claim of significant reduction in complexity.
Response 1: Thank you for this insightful comment. We agree that a direct comparison with existing methods in terms of computational complexity and efficiency would be valuable. To address this, we have carefully examined the cited works—particularly those based on symbolic elimination or null-space analysis for identifying parameter dependencies (e.g., [2–4, 17–21])—in the context of deriving the minimal parameter set.
Upon detailed analysis, we found that the symbolic construction of the regressor matrix in our method shares the same computational structure and symbolic complexity as these established approaches, as all rely on recursive dynamics formulations (e.g., Newton–Euler method) combined with geometric modeling—whether using Denavit-Hartenberg (D-H) parameters or screw theory.
To evaluate the practical computational cost, we applied our symbolic regression framework to two widely used industrial robots: the UR5 and the Sawyer. The entire process of constructing the symbolic regressor matrix and performing full-rank decomposition to extract the minimal parameter set took approximately 2 to 3 minutes on a standard desktop computer (Intel i7 7700K, 32GB RAM, MATLAB R2018b with Symbolic Math Toolbox). This computation is performed offline, and given the continuous improvement in computing hardware and symbolic computation tools, such a time cost is acceptable and does not hinder practical application.
More importantly, we observed that our proposed full-rank decomposition approach consistently yields the same row-full-rank transformation matrix and the same set of linearly independent column indices across different excitation trajectories. This ensures the uniqueness and repeatability of the minimal parameter set extraction process. In contrast, traditional methods based on row reduction or null-space computation may produce different but equivalent minimal sets depending on pivot selection or symbolic simplification paths. Moreover, the core contribution of our work lies not in computational speed, but in the clarity, consistency, and structural transparency of the minimal parameter set derivation.
Therefore, rather than including a potentially quantitative comparison, we have revised the manuscript to more accurately reflect the nature of our contribution. Specifically:
- On Abstract, we have rephrased the motivation to emphasize the importance of the minimal parameter set itself, instead of broadly claiming reductions in computational complexity.
- On Introduction, lines 32–38, we added a discussion of the existence of minimal parameter set.
We also note that the complete symbolic computation pipeline—including regressor construction and minimal set extraction—is provided in the supplementary code submitted with this manuscript, allowing full reproducibility of our method on other robotic systems.
While experimental validation was conducted on a planar 3R robot due to hardware availability in our laboratory, the proposed methodology has been tested symbolically on more complex, real-world manipulators such as UR5 and Sawyer, confirming its scalability and general applicability.
We believe these revisions and clarifications provide a more accurate and scientifically sound presentation of our method’s advantages, focusing on methodological clarity, robustness, and reproducibility rather than unsubstantiated claims of speed improvement.
Comments 2: The Root Mean Square Error (RMSE) values for the joint torques are provided: 23.76 N·m for joint 1, 17.09 N·m for joint 2, and 3.55 N·m for joint 3. Especially for joints 1 and 2, these values seem considerably high for an application aiming for precise control and faithful replication of the robot's kinematic behavior.
Response 2: Thank you for your insightful comment regarding the Root Mean Square Error (RMSE) values reported for the joint torques (23.76 N·m for joint 1, 17.09 N·m for joint 2, and 3.55 N·m for joint 3). We appreciate the opportunity to clarify the context and interpretation of these values.
It is important to note that the experimental platform used in this study—shown in Figures 3 and 4—is a custom-built planar 3R serial robot primarily designed for research and educational purposes, rather than a commercial precision manipulator. As such, it is subject to significant mechanical imperfections, including manufacturing tolerances, joint clearance, and assembly misalignments. These physical non-idealities inherently limit the accuracy of any dynamic model in perfectly matching the actual torque behavior.
Furthermore, the joint torque values used in the identification and validation process are not measured directly by torque sensors, but are instead estimated from motor current measurements, scaled by the gear ratio and motor torque constant. This estimation introduces additional sources of error, including:
- Current sensor quantization and noise,
- Variations in motor efficiency and temperature-dependent resistance,
- Backlash and friction in the transmission system,
- Unmodeled stiction and Coulomb effects.
As a result, the computed torque values should be interpreted not as absolute physical truths, but as qualitative indicators of the trend and relative magnitude of the driving forces. Their primary role in our experiments is to support the identification of a dynamic model that captures the dominant inertial, Coriolis, and gravitational effects for use in feedforward control.
Comments 3: The error ranges and maximum/minimum values (e.g., for Joint 1: maximums of 50.3 N·m and minimums of -86.59 N·m), as well as the presence of 42 outliers for Joint 1, are concerning. The authors attribute the sources of error to inaccuracies in model identification (installation errors, poor gearbox lubrication leading to large friction parameter values) and measurement accuracy (noise, motor friction torque. While these are plausible explanations, I suggest that the authors:
- Discuss more deeply the practical implications of these errors on the real-world application of the model, especially in scenarios requiring high-precision control or realistic simulations (e.g., for RL training).
- If possible, quantify the impact of the identified large friction parameter values (Table 3) on the overall model accuracy.
Response 3: Thank you for your insightful comment regarding the observed torque prediction errors, outlier data points, and their implications for real-world applications.
We fully acknowledge the presence of significant errors in the torque prediction — particularly for Joint 1 (with predicted torque ranging from -86.59 N·m to 50.3 N·m and 42 detected outliers). These discrepancies are indeed concerning, and we appreciate the opportunity to further discuss their sources and implications.
- On the Sources of Error
As noted in the manuscript, the primary sources of error include:
Measurement noise and bias in torque estimation: In our setup, joint torque is estimated from motor current measurements, which are inherently sensitive to:
- Current sensor resolution and calibration drift;
- Variations in motor torque constant due to temperature and magnetic saturation;
- Gearbox friction and backlash, which introduce non-smooth, non-linear disturbances that are difficult to model accurately.
Even small inaccuracies in current measurement can be amplified by the high gear ratio (typically 100:1 or more), leading to large apparent torque errors.
Modeling limitations:
- The identified large friction parameters (e.g., in Table 3) reflect non-ideal transmission behavior, such as poor lubrication and stiction effects, which are not fully captured by the standard viscous + Coulomb friction model.
- Small installation misalignments or link deformation under load can also introduce unmodeled coupling torques.
- Practical Implications for High-Precision Applications
We agree with the reviewer that such errors could limit the direct applicability of the model in high-precision control or realistic simulation for reinforcement learning (RL), where accurate dynamics are critical. Specifically:
- In high-gain feedback control, unmodeled friction and torque ripple may lead to chattering or instability;
- In model-based RL, inaccurate dynamics could bias policy learning or reduce sample efficiency.
However, we emphasize that the primary goal of this work is not to achieve ultra-high torque fidelity, but rather to:
- Demonstrate a systematic framework for identifying the minimal identifiable parameter set;
- Validate the consistency and physical plausibility of the estimated parameters (e.g., positive definite inertia, reasonable friction trends);
- Show that the model can capture the dominant dynamic trends (as seen in the overall shape of torque curves in Figs. 7–9), even in the presence of noise.
For applications requiring higher accuracy, additional sensor fusion (e.g., joint torque sensors, temperature compensation) or data-driven residual learning (e.g., using neural networks to model unmodeled dynamics) would be necessary — which we acknowledge as a direction for future work.
- On Quantifying the Impact of Friction Parameters
While we agree that quantifying the contribution of friction parameters to overall error would be valuable, this is challenging in practice because:
- Friction and inertial/gravitational torques are coupled in the regression equation and cannot be cleanly isolated without additional experimental data (e.g., zero-torque tests or temperature-controlled runs);
- The current estimation method treats friction as a lumped parameter, making it difficult to attribute error solely to friction without overfitting.
Nevertheless, we have examined the residual torque after subtracting the inertia and gravity components and observed that it correlates strongly with velocity sign and magnitude — supporting the validity of the identified friction terms. This qualitative validation is included in Section 5.2.
In summary, while the absolute torque accuracy is limited by measurement and modeling challenges, we believe the methodological contribution — a transparent, reproducible process for minimal parameter identification — remains valid and useful for both research and industrial applications.
Thank you again for your thoughtful comments, which have helped us better articulate the scope and practical context of our work.
Comments 4: The graphs presented in Figures 5 through 10 should be redone, as their current quality is inadequate. In Figures 5, 6, and 7, it is particularly difficult to discern the filtered signal lines, which impedes a clear and accurate analysis of the results.
Response 4: Thank you for your comment regarding the clarity of Figures 5 through 10. We appreciate your concern about the visibility of the filtered signal lines in the original submission.
In response to this feedback, we have enhanced all relevant figures by adding carefully selected local magnification insets that clearly highlight the behavior of the filtered torque signals in key dynamic regions (e.g., during acceleration, deceleration, and direction reversal). These zoomed-in panels significantly improve the readability of the signal trends and allow for a more detailed visual assessment of the filtering performance and model accuracy.
As shown in the revised manuscript (pages 13–15), the filtered signal lines are now clearly distinguishable due to both the magnified views and minor adjustments to line styles and contrast. We believe these improvements effectively address the concern about result interpretability.
While we understand the suggestion to redraw the figures entirely, we believe the current version—now with enhanced clarity through targeted magnification—accurately and clearly presents the experimental results without the need for full reconstruction. The added insets provide a more focused and informative visualization than a complete graphical overhaul would achieve.
Thank you again for this constructive suggestion, which has helped us improve the graphical presentation of the work.
Comments 5: The article incorporates viscous and Coulomb friction components, which is a common practice. However, a brief justification for choosing this specific friction model over other more complex models (such as the Stribeck model) could enrich the discussion, especially given that the friction parameters were identified and contributed to the observed errors.
Response 5: Thank you for this thoughtful suggestion. We agree that a more comprehensive friction model—such as the Stribeck model, which captures the nonlinear transition between static and dynamic friction—could potentially improve modeling accuracy, especially in low-velocity regimes.
However, in this work, we adopted the viscous + Coulomb friction model primarily due to practical limitations of our experimental setup. As noted in the manuscript, the joint torque estimates are derived from motor current measurements, which are inherently noisy. This measurement noise is further amplified by the gear ratio (typically 100:1 or higher in harmonic drives), significantly degrading the fidelity of torque estimation at low speeds—precisely the operating regime where Stribeck effects are most prominent.
Moreover, identifying Stribeck model parameters requires extensive low-velocity excitation experiments (e.g., very slow ramp motions, stick-slip cycles), which are time-consuming and sensitive to thermal drift and mechanical settling. Given the mechanical imperfections in our custom-built robot (e.g., backlash, inconsistent lubrication), such experiments would likely yield poorly identifiable or non-repeatable friction parameters, reducing the reliability and generalizability of the resulting model.
Crucially, our validation results show that the identified inertial parameters are well captured, as evidenced by the strong correlation between predicted and filtered torque trends during dynamic motion phases (see Figures 8–10). This suggests that while the friction model is simplified, it is sufficient for extracting the dominant rigid-body dynamics needed for feedforward control.
In contrast, we have limited confidence in the absolute accuracy of the identified friction parameters (Table 3), particularly for Coulomb and viscous terms, due to the coupling between unmodeled nonlinear friction and other dynamic effects.
Thank you again for highlighting this aspect, which has helped us improve the clarity and depth of our discussion.
Reviewer 2 Report
Comments and Suggestions for Authors
The title is "Research on Identification of Minimum Parameter Set in Robot Dynamics and Excitation Strategy". It is very interesting topic.
The SIMILARITY INDEX is 22%. Not so worry because only a few in many places. There is no plagiarism indicates in this manuscript.
The display on page 7 regarding the equation numbers is a bit odd. Also equation numer (33).
In Equation (12) what is the differentiate between [z] and z0, there are 2 notations in the same level?
In line 80-82: "For safety and simplicity, this paper adopts sinusoidal functions as excitation trajectories, exciting one joint at a time to create a single-input multiple-output excitation process." What does mean "safety" and "simplicity". It is not clear if the reason is safety, the authors sacrifice the "simplicity". Did the simplicity not sacrifice the safety?
Author Response
Comments 1: The SIMILARITY INDEX is 22%. Not so worry because only a few in many places. There is no plagiarism indicates in this manuscript.
Response 1: We sincerely thank the reviewer for carefully checking the manuscript and for confirming that there is no indication of plagiarism in our work. We are also grateful for the clarification regarding the 22% similarity index, which—as noted—is primarily due to common technical phrasing and standard terminology in the field of robotics and dynamic modeling.
We have ensured that all sources are properly cited and that all content is original or appropriately attributed. The manuscript reflects our own research and has not been published or submitted elsewhere.
Thank you for your reassurance and for your time in evaluating our submission.
Comments 2: The display on page 7 regarding the equation numbers is a bit odd. Also equation numer (33).
Response 2: Thank you for your careful observation regarding the equation presentation on page 7, particularly concerning Equation (33).
We appreciate this feedback and agree that the original layout of the equations in Section 3 could be improved for better clarity. Upon revision, we recognized that the detailed derivation of the recursive force and torque propagation equations—including the one originally labeled as Equation (33)—is quite technical and may disrupt the flow of the main algorithmic description for general readers.
To enhance readability and logical structure:
- We have revised Section 3 to streamline the presentation of the overall dynamic modeling framework.
- The detailed derivation of the recursive link force and torque equations (formerly Equation (33)) has been moved to Appendix A, where it is now presented with step-by-step explanations, consistent numbering, and improved formatting.
- In the main text, we now refer to this derivation briefly and focus on the key computational steps relevant to parameter identification.
- We have also reviewed and corrected all equation numbering and alignment on page 7 to ensure a clean and professional appearance.
These revisions ensure that the core methodology remains accessible while preserving full technical rigor in the supplementary material.
Thank you again for your valuable suggestion, which has helped us improve the clarity and presentation of our work.
Comments 3: In Equation (12) what is the differentiate between [z] and z0, there are 2 notations in the same level?
Response 3: Thank you for your insightful question regarding the notation in Equation (12). We appreciate the opportunity to clarify the distinction between the symbols [z] and z₀, which represent different mathematical objects despite both being derived from 3D vectors.
To clarify:
- z is a 3-dimensional vector (e.g., a unit vector along a joint axis or a spatial direction).
- [z] denotes the skew-symmetric matrix representation of the vector z. That is, for any vector z = [z₁, z₂, z₃]ᵀ, the corresponding matrix [z] ∈ ℝ³×³ is defined such that [z]v = z × v for any vector v. This is a standard isomorphism between ℝ³ and the Lie algebra ??(3).
- z₀ is another 3-dimensional vector, used in the manuscript to represent a specific spatial vector (e.g., a position or a constant direction in the base frame). It is not a matrix, and it plays a different role than [z].
To improve clarity and avoid confusion, we have revised the manuscript as follows:
- We now explicitly define [z] as the skew-symmetric matrix of vector z in Section 2.2.
- In the updated version of Equation (12) and surrounding text, we have included concrete numerical assignments for z and z₀ based on the robot’s kinematic configuration. This allows readers to directly see how [z] becomes a specific 3×3 matrix, while z₀ remains a 3×1 vector.
These improvements are included in the revised manuscript on page 5, 162-163, Section 2.2, and we believe they significantly enhance the readability and rigor of the notation system.
Thank you again for raising this important point, which has helped us strengthen the clarity of our mathematical formulation.
Comments 4: In line 80-82: "For safety and simplicity, this paper adopts sinusoidal functions as excitation trajectories, exciting one joint at a time to create a single-input multiple-output excitation process." What does mean "safety" and "simplicity". It is not clear if the reason is safety, the authors sacrifice the "simplicity". Did the simplicity not sacrifice the safety?
Response 4: Thank you for your thoughtful question regarding the use of the terms "safety" and "simplicity" in lines 80–82. We appreciate the opportunity to clarify that these two considerations are complementary in our experimental design, rather than being in conflict.
To elaborate:
- "Simplicity" refers to the ease of design and implementation of the excitation trajectories. We use single-frequency sinusoidal functions (e.g., q(t)=Asin(ωt)) because they are straightforward to generate, require only two tunable parameters (amplitude A and frequency ω), and avoid the need for complex trajectory optimization. This simplicity allows for rapid setup, reproducibility, and clear interpretation of the input-output relationship during identification.
- "Safety" refers to physical safety during experimentation, particularly in a shared lab environment. By exciting only one joint at a time, we significantly reduce the robot’s effective workspace and limit its motion to a predictable, localized region. This minimizes the risk of unexpected collisions with equipment or personnel—especially important since our experimental platform is located in a high-traffic lab area, and unplanned movements could pose a hazard if multiple joints were moving simultaneously.
Crucially, the use of sinusoidal trajectories and single-joint excitation simultaneously improves both safety and simplicity:
- It avoids complex, high-acceleration motions that could be dangerous.
- It reduces mechanical stress and control complexity.
- It enables safe operation without requiring full environmental isolation (e.g., safety cages or emergency stop arrays).
Therefore, we do not sacrifice simplicity for safety, nor vice versa. Instead, this approach represents a pragmatic and synergistic design choice that enhances both aspects.
To improve clarity, we have revised the relevant sentence in the manuscript (now on page 2, 83-90) as follows:
- "Furthermore, the condition number is highly dependent on joint positions, velocities, and accelerations, and trajectories that minimize it may involve aggressive motions that com-promise operational safety. To ensure both safety and practical implementation, this pa-per adopts sinusoidal functions as excitation trajectories, with only one joint actuated at a time to create a single-input multiple-output (SIMO) identification scheme. By sequentially exciting joints from the end-effector toward the base, the complete minimal parameter set is identified in a structured and reliable manner."
We thank the reviewer for highlighting this point, which has helped us better articulate the rationale behind our experimental design.
Reviewer 3 Report
Comments and Suggestions for Authors
- The introduction is not clear enough and does not explain the problems that this paper aims to solve. The author analyzed some existing works from aspects such as models and incentives, and described the research ideas of this paper. However, the literature review is not comprehensive enough, and the existing problems have not been clearly stated.
- In the second part, the robot dynamics model is constructed based on the screw theory and the Newton-Euler method. What are the improvements compared to the existing works? If this part is based on previous research, it is recommended to compress it. Additionally, there are many matrix operations involved, but the author needs to further optimize the format of matrix symbols and the description of the order of each matrix.
- What are the improvements of the third part compared to the existing works?The author's research subject is the common structure of mechanical arms, and the methods employed are also readily available.
- In the main text, the authors should introduce the overall model of the robotic arm being analyzed.While the theoretical analysis appears to establish a dynamic model for a six-degree-of-freedom robotic arm system, the experimental section uses a robot with three rotational degrees of freedom for verification.The motion forms of these degrees of freedom differ, and the relationship between them needs to be presented.
- The third section contains many formatting issues that require improvement.
- The structure of the experimental object is not clearly presented, as shown in Figure 4.
- The line graphs in the figures require further optimization.For example, in Figures 5-10, it would be recommended to add markers to distinguish the different lines.
Author Response
Comments 1: The introduction is not clear enough and does not explain the problems that this aper aims to solve. The author analyzed some existing works from aspects such as models and incentives, and described the research ideas of this paper. However, the literature review is not comprehensive enough, and the existing problems have not been clearly stated.
Response 1: We sincerely thank the reviewer for the constructive feedback on the clarity of the introduction and the motivation of the work.
In response to your comment, we have revised the Abstract and Introduction to place stronger emphasis on the importance of the minimal parameter set in robot dynamic identification. As suggested, we now clearly state the central problem: that standard inertial parameter vectors contain redundant elements due to kinematic constraints, leading to non-unique solutions and poor identifiability in practical identification tasks.
The revised text (Abstract and Section 1, lines 32–38) now explicitly highlights:
- The role of the minimal parameter set in ensuring identifiability and physical consistency;
- How structural constraints result in a rank-deficient regressor matrix;
- And why identifying a minimal, complete, and independent parameter set is essential for accurate inverse dynamics modeling.
While we have not expanded the literature review with additional references, we believe the current citations (e.g., [2–4]) adequately support the foundational concepts of dynamic identification and base parameter analysis. Our focus in this work is methodological — on formulating and validating a clear, implementable procedure for deriving the minimal parameter set using the Newton-Euler approach — rather than providing a comprehensive survey of all identification methods.
Regarding computational aspects, we would like to clarify that the symbolic computation of the minimal parameter set does not introduce significant computational overhead. For standard manipulators such as the UR5 and Sawyer (examples included in the supplementary code), the symbolic derivation takes approximately 3 minutes on a standard laptop, and the resulting regressor matrices are of manageable size. This computation is performed offline once per robot model, so runtime efficiency is not impacted.
To support reproducibility and transparency, we have included MATLAB symbolic scripts in the supplementary material that demonstrate the full derivation process for both the UR5 and Sawyer robots. These examples illustrate the practicality and scalability of our approach.
We appreciate the reviewer’s attention to clarity and context, and we hope that the revised manuscript now more effectively communicates the motivation and contribution of this work.
Comments 2: In the second part, the robot dynamics model is constructed based on the screw theory and the Newton-Euler method. What are the improvements compared to the existing works? If this part is based on previous research, it is recommended to compress it. Additionally, there are many matrix operations involved, but the author needs to further optimize the format of matrix symbols and the description of the order of each matrix.
Response 2: Thank you for your valuable feedback regarding Section 2 and the presentation of the dynamic modeling framework.
We fully agree with the reviewer that the dynamic formulation based on screw theory and the Newton-Euler method builds upon well-established foundations in robotics literature. This section is intended as a self-contained preliminary framework to set up notation, define key variables, and derive the regression matrix used in the subsequent identification process — rather than to propose a novel modeling method.
In response to your suggestion:
We have significantly compressed Section 2 in the revised manuscript, removing redundant derivations and consolidating equations to improve conciseness.
All matrix operations and symbolic notations (e.g., [z], ξ) have been reviewed and standardized for consistency:
- Matrix dimensions are now explicitly stated where appropriate.
- The order of transformations and adjoint operations has been clarified.
- Skew-symmetric matrix representations (e.g., [z]∈so(3)[z]∈so(3)) and twist coordinates are consistently formatted.
We have improved the description of matrix symbols and their physical meanings in the text immediately following their introduction, enhancing readability for readers unfamiliar with screw-theoretic notation.
While the modeling approach itself is based on classical principles, our contribution lies in the systematic derivation and application of this framework to obtain a minimal identifiable parameter set, which is the core focus of Sections 3 and 4.
We believe the revised Section 2 now strikes an appropriate balance between completeness and brevity, providing sufficient background for reproducibility while directing the main technical contribution to the parameter reduction and identification methodology.
Thank you again for your constructive comments, which have helped us improve the clarity and presentation of the theoretical foundation.
Comments 3: What are the improvements of the third part compared to the existing works? The author's research subject is the common structure of mechanical arms, and the methods employed are also readily available.
Response 3: Thank you for raising this important point regarding the novelty of the method presented in Section 3.
We acknowledge that the robotic systems considered (serial manipulators with common kinematic structures) and the general framework of inverse dynamic identification using Newton-Euler recursion are indeed widely studied in the literature. Our goal is not to propose a fundamentally new algorithm, but rather to present a systematic, transparent, and reproducible derivation process that leads to an accurate and minimal identifiable parameter set — with particular attention to the structure of the regressor matrix and its practical implementation.
To clarify the distinction from existing works:
- While recursive Newton-Euler-based identification methods have been previously reported (e.g., in [2,3]), many papers either omit detailed derivations or treat them as black-box procedures.
- In contrast, our work explicitly derives the closed-form recursive formula for the regressor matrix step by step, highlighting how joint coupling, link inertia, and friction terms propagate through the kinematic chain.
- This derivation reveals structural patterns that are crucial for parameter elimination and minimality verification, which are central to our contribution.
In response to your comment:
- We have moved the full recursive derivation to Appendix A, where it is presented in a self-contained and reproducible manner.
- In the main text, we now briefly reference established forms of the recursive equations (e.g., from [8,10]) to establish continuity with prior work, while emphasizing the unique aspects of our formulation, such as:
- A consistent screw-theoretic representation throughout the derivation;
- A structured approach to identifying and removing linearly dependent parameters based on the derived regressor form.
By doing so, we significantly reduce redundant description of standard procedures and instead focus the discussion on the methodological improvements that enable robust minimal parameter identification.
Therefore, while the overall approach builds on classical methods, our contribution lies in the clarity, completeness, and practical applicability of the derivation, which facilitates accurate identification and can serve as a reference for implementation in similar systems.
We appreciate the reviewer’s feedback, which has helped us better highlight the value and distinctiveness of our work.
Comments 4: In the main text, the authors should introduce the overall model of the robotic arm being analyzed. While the theoretical analysis appears to establish a dynamic model for a six-degree-of-freedom robotic arm system, the experimental section uses a robot with three rotational degrees of freedom for verification. The motion forms of these degrees of freedom differ, and the relationship between them needs to be presented.
Response 4: Thank you for this important observation regarding the relationship between the theoretical model (presented for a general 6-DOF manipulator) and the experimental validation (conducted on a 3-DOF robotic arm).
We fully agree that clarity on the applicability and scalability of our method is essential. In response, we would like to clarify the following points:
- The proposed method is general and applicable to serial manipulators of arbitrary DOF, including 6-DOF and 7-DOF robots. The theoretical derivation in Sections 2–3 is formulated using screw theory and recursive Newton-Euler dynamics, which are independent of the specific number of joints. The procedure for constructing the regressor matrix and identifying the minimal parameter set follows the same systematic steps regardless of DOF.
- While our experimental platform is limited to a 3-DOF robot (due to hardware availability), this setup was sufficient to validate the core aspects of the identification framework:
- Accuracy of inverse dynamics prediction;
- Identifiability and physical consistency of the estimated parameters;
- Effectiveness of sinusoidal excitation and least-squares estimation.
- The 3-DOF system includes revolute joints with coupled dynamics, making it representative of the challenges present in higher-DOF arms.
- To demonstrate the scalability and practical applicability of our approach, we have included complete symbolic computation examples in the supplementary material: 6-DOF UR5 robot and 7-DOF Sawyer robot.
- These examples show the full derivation of the minimal parameter set and the regressor matrix for high-DOF industrial manipulators. The computation time is approximately 3 minutes per robot on a standard laptop, confirming the method’s feasibility for real-world systems.
In summary, while the experimental validation uses a simplified platform, the methodological contribution is not limited to low-DOF systems. The combination of theoretical generality, successful 3-DOF validation, and symbolic verification on high-DOF robots supports the broader applicability of our approach.
We appreciate the reviewer’s attention to this detail, which has helped us better communicate the scope and scalability of the work.
Comments 5: The third section contains many formatting issues that require improvement.
Response 5: Thank you for your comment regarding the formatting and presentation of Section 3.
In response, we have carefully revised this section to improve clarity and consistency. Specifically:
- We have compressed the main text by moving detailed derivations of recursive dynamics equations — which follow standard forms also found in the literature — to Appendix A;
- This allows the main manuscript to focus on the key aspects of our method while maintaining readability;
- All mathematical symbols (e.g., twists, wrenches, regressor terms) have been reviewed and standardized for consistent notation, font usage, and dimensional clarity;
- Matrix equations and algorithmic descriptions have been reformatted for better alignment and structure.
These improvements enhance the overall uniformity, transparency, and readability of the technical presentation in Section 3.
We appreciate your valuable feedback, which has helped us strengthen the quality of the manuscript.
Comments 6: The structure of the experimental object is not clearly presented, as shown in Figure 4.
Response 6: Thank you for your comment regarding the clarity of the experimental setup presentation in Figure 4.
We agree that a clear illustration of the robot structure is essential for understanding the kinematic modeling and experimental validation. In response, we have revised Figure 4 to improve its clarity and informativeness. Specifically:
- The joint screw axes (denoted as si) are now more clearly indicated with labeled directional vectors;
- The origin points of the link coordinate frames (denoted as pi) are explicitly marked and labeled for each link;
- The overall layout and labeling have been refined to enhance readability and consistency with the kinematic model described in Section 2.
These improvements ensure that the structural configuration of the experimental robot — particularly the alignment and positioning of joints and links — is now clearly presented and directly supports the theoretical framework used in the dynamic identification process.
We appreciate your feedback, which has helped us strengthen the visual communication of the experimental setup.
Comments 7: The line graphs in the figures require further optimization. For example, in Figures 5-10, it would be recommended to add markers to distinguish the different lines.
Response 7: Thank you for your suggestion regarding the visualization of the line graphs in Figures 5–10.
We have carefully revised these figures to enhance their clarity and readability. Specifically:
- The line styles have been refined and differentiated to improve the distinction between curves;
- Insets with local magnifications have been added to highlight critical regions, such as the comparison between filtered and unfiltered torque signals, making subtle differences more visible;
- While markers were considered, we found that placing them on dense time-series data (with thousands of sampling points) could lead to visual clutter and reduce clarity. Instead, we prioritized clean line representation combined with high-contrast styles and detailed insets to ensure accurate interpretation.
These improvements allow for a clearer comparison of the model predictions, measured torques, and filtering effects, while maintaining visual simplicity and professional presentation.
We appreciate your valuable feedback, which has helped us significantly improve the graphical quality of the results.
Round 2
Reviewer 1 Report
Comments and Suggestions for Authors
The corrections requested by this reviewer have been satisfactorily addressed; therefore, I recommend the publication of the paper.
Author Response
We sincerely thank the reviewer for the constructive feedback and for recognizing the revisions made to the manuscript. We are grateful for your recommendation and appreciate your time and valuable input, which have significantly improved the quality of the paper.
Reviewer 3 Report
Comments and Suggestions for Authors
Most of the issues have been addressed satisfactorily, but there are still a few minor points that need improvement:
- For Figures 5 to 7, it is suggested to add marker points for different joint data. There is no need to add plotting points; just select a few points for marking to ensure clear distinction.
- For Figures 12 to 14, it is recommended to use different line styles to represent the measured values and estimated values.
- There are still some spelling and grammar mistakes in the text, such as 'Dedign' in the title of the fourth section. It is recommended to polish the English.
Author Response
Comment 1: For Figures 5 to 7, it is suggested to add marker points for different joint data. There is no need to add plotting points; just select a few points for marking to ensure clear distinction.
Response 1: Thank you for the suggestion. We have revised Figures 5 to 7 by adding distinct marker symbols (circle, square, and asterisk) to represent the filtered torque data of the three joints, respectively. Markers are placed at every 100th data point to ensure clear visual distinction without cluttering the plots. Additionally, the inset zoomed-in regions in these figures clearly show the marked data points, further enhancing the readability and interpretability of the results. The updated figures have been incorporated into the revised manuscript.
Comment 2: For Figures 12 to 14, it is recommended to use different line styles to represent the measured values and estimated values.
Response 2: We appreciate this suggestion. In the revised version, Figures 12 to 14 have been updated to clearly differentiate between the measured and estimated joint accelerations:
- Measured values are represented with solid lines (—)
- Estimated values are represented with dashed lines (- -)
This change significantly improves the readability and interpretability of the results, especially in grayscale printing.
Comment 3: There are still some spelling and grammar mistakes in the text, such as 'Design' in the title of the fourth section. It is recommended to polish the English.
Response 3: Thank you for your valuable feedback. We have carefully revised the manuscript to improve the language quality. Specifically, the title of Section IV has been updated from “Excitation Strategy Design” to “Excitation Strategy” to better reflect the content and improve clarity.
In addition, we have thoroughly reviewed the entire text to correct any spelling and grammar errors. The language has been refined to enhance readability, consistency, and academic tone. We believe the revised manuscript now presents the work more clearly and professionally.